# Cognitive Performance in Relation to Systemic and Brain Iron at Perimenopause

**DOI:** 10.3390/nu17050745

**Published:** 2025-02-20

**Authors:** Amy L. Barnett, Michael J. Wenger, Pamela Miles, Dee Wu, Zitha Redempta Isingizwe, Doris M. Benbrook, Han Yuan

**Affiliations:** 1Psychology and Cellular and Behavioral Neurobiology, The University of Oklahoma, Norman, OK 73019, USA; amy.l.barnett029@gmail.com; 2Obstetrics and Gynecology, College of Medicine, The University of Oklahoma Health Sciences Center, Oklahoma City, OK 73104, USA; pamela-miles@ouhsc.edu; 3Radiological Sciences, College of Medicine, The University of Oklahoma Health Sciences Center, Oklahoma City, OK 73104, USA; dee-wu@ouhsc.edu; 4Gynecological Oncology, College of Medicine, The University of Oklahoma Health Sciences Center, Oklahoma City, OK 73104, USA; zitha-isingizwe@ouhsc.edu (Z.R.I.); doris-benbrook@ouhsc.edu (D.M.B.); 5Stephenson School of Biomedical Engineering, The University of Oklahoma, Norman, OK 73019, USA; hanyuan@ou.edu

**Keywords:** iron, perimenopause, memory, attention, aging

## Abstract

Background: The literature on the relationships among blood iron levels, cognitive performance, and brain iron levels specific to women at the menopausal transition is ambiguous at best. The need to better understand these potential relationships in women for whom monthly blood loss (and thus iron loss) is ceasing is highlighted by iron’s accumulation in brain tissue over time, thought to be a factor in the development of neurodegenerative disease. Methods: Non-anemic women who were either low in iron or had normal iron levels for their age and race/ethnicity provided blood samples, underwent MRI scans to estimate brain iron levels, and performed a set of cognitive tasks with concurrent EEG. Results: Cognitive performance and brain dynamics were positively related to iron levels, including measures associated with oxygen transport. There were no relationships between any of the blood measures of iron and brain iron. Conclusions: Higher iron status was associated with better cognitive performance in a sample of women who were neither iron deficient nor anemic, without there being any indication that higher levels of systemic iron were related to higher levels of brain iron. Consequently, addressing low iron levels at the menopausal transition may be a candidate approach for alleviating the “brain fog” commonly experienced at menopause.

## 1. Introduction

Much of what is known about the role of iron in health generally and brain function and cognition in humans has come from studies of iron deficiency in infants, adolescents, and women of reproductive age. Iron deficiency is the most common cause of anemia, with more than one billion cases reported worldwide in recent years [1]. Based on hemoglobin (Hb) levels, 40% percent of children (aged 6–59 months), 30% of women who are not pregnant, and 37% of women who are pregnant are anemic (Hb < 110 g/L) [1]; these rates may be even higher if criteria for erythropoiesis are considered (Hb < 120 g/L) [2,3].

In the brain, as in all tissues in the body, iron is present in each cell. Iron is required for many functions in the brain, including the production of myelin, neurogenesis and synaptogenesis, neurotransmitter synthesis and re-uptake, and oxygen transport [4,5,6,7,8,9,10]. Later in life, the accumulation of non-heme iron in brain tissue represents a potential source of oxidative stress [11,12], which may be prodromal to neurodegenerative diseases, including Alzheimer’s disease.

Iron is notably important in the synthesis and synaptic regulation of a variety of neurotransmitters, with dopamine (DA) being the most widely studied [13,14,15,16,17,18]. Iron is a cofactor for tyrosine hydroxylase, which is an enzymatic precursor to DA [6,19]. Iron deficiency can lead to a down-regulation of DA D1 and D2 receptors and alters the functioning of the DA transporter [7,9,20,21,22] such that the concentration of extracellular DA increases. Iron is also critical in the synthesis and regulation of serotonin, though this has received far less attention [4].

### 1.1. Iron, Aging, and the Menopausal Transition

Women of reproductive age are more likely to be affected by iron deficiency than men due to monthly blood loss. In men, given that there is no monthly blood loss, iron accumulates in tissues, including brain tissue, across the lifespan. For women, this changes at the menopausal transition, when iron is no longer being lost, and iron accumulation in the brain has the potential to increase or even accelerate after the cessation of menstruation. This age-related accumulation of iron has motivated the oxidative stress theory of aging, which argues that iron accumulation in the brain increases with age and can induce the production of highly reactive hydroxyl radicals, which increases the risk of oxidative stress [11]. Reactive oxygen species are formed when ${\mathrm{Fe}}^{2+}$ reacts with the hydrogen peroxide, producing hydroxyl radicals that can directly damage DNA [23,24,25]. In turn, oxidative stress contributes to cognitive decline and neurodegenerative diseases such as Alzheimer’s disease (AD) and Parkinson’s disease [26,27,28].

Age is positively correlated with increasing deposits of iron in the basal ganglia, putamen, globus pallidus, and substantia nigra [12,23,29,30,31], and increasing levels of iron deposition can be observed across the entire lifespan. Disruptions in iron homeostasis during aging may be due to other forms of iron, such as neuromelanin [25,32]. Neuromelanin-containing organelles have high concentrations of iron and are the dominant form of iron in catecholaminergic neurons [24]. However, although it protects neurons from toxicity, neuromelanin may sequester excess iron prompting neurodegeneration. Leakage of neuromelanin from these damaged neurons can induce cell death by inducing microglial activation, which, in turn, releases more neuromelanin, exacerbating the process as a negative feedback loop [23]. Similarly, iron toxicity activates glial cells, which then also dysregulates hepcidin (a peptide that regulates iron homeostasis), therefore causing a “self-propelling cycle” of neurodegeneration [24].

The relationship between neurodegeneration, cognitive decline, and systemic (blood) levels of iron as a function of dietary iron is quite ambiguous [33]. A meta-analysis by Hosking et al. [34] found conflicting results with respect to systemic iron status, aging, and neurodegeneration across six cohort studies. An analysis of a National Health and Nutrition Examination Survey (NHANES) dataset found that high levels of both transferrin saturation and cholesterol were associated with an increased risk for AD, but no such association was found with high transferrin saturation alone [35]. A longitudinal cohort study in Australia found a higher risk of mild cognitive impairment (MCI) with higher levels of iron intake [36]. In that Australian cohort, lower iron intake was associated with reduced risk for MCI in women but with higher risk in men, even when taking oral supplements. Alternatively, in the Alzheimer’s Disease Neuroimaging Initiative cohort, Ayton et al. [37] found that higher levels of serum ferritin (sFt) were associated with an increased risk in cognitive decline, but this may be limited to individuals who carry the APOE-ε4 allele [34].

The ambiguity regarding the potential relationships among systemic iron, brain iron, brain function, cognitive decline, and risk for neurodegenerative diseases increases when the focus is placed on women at the menopausal transition, particularly given the sparsity of the relevant literature. To the best of our knowledge, there exist only two studies that address iron levels and cognition in the time period encompassing the menopausal transition [38,39]. These studies provide conflicting evidence with respect to iron status, and neither provides data on brain iron levels. Lam et al. [38] found evidence for an inverse relationship between iron status and cognition after menopause. This work examined the sex-specific associations of plasma concentrations of iron, copper, and zinc with cognitive function in older adults (>60 y) and found that higher levels of iron in women (mean age 74.7 y) were associated with poorer scores on short-term and long-term measures of memory. In contrast, Andreeva et al. [39], using the SU.VI.MAX 2 cohort [40], found extremely limited evidence for a negative relationship between iron status (as measured using sFt at baseline) and cognition, as measured by the forward digit span task, and this effect was quite weak, such that each unit increase in sFt was associated with just slightly more than a one-tenth of one unit decrease in performance. For all other measures, there was no relationship between iron status (measured using either Hb and sFt) at the beginning of the six-year study period, change in iron status over six years, and cognitive performance at six years, as assessed using tasks that included cued recall, phonemic fluency, semantic fluency, forward and backward digit span, and trail-making tests.

### 1.2. Purpose of the Present Study

Unfortunately, there are no data on changes in brain iron accumulation specifically among women at perimenopause and their potential relationship to changes in cognition and brain function. In addition, studies that have examined iron and cognition in this age range either report only blood measures [38,39] or report only brain measures [12,29,31], with all studies dedicated to the latter being focused on brain structure (using MRI) rather than brain function (as is possible with EEG), and with none of these separating out results by biological sex. None of the extant studies report on behavior, brain function, and brain structure. Thus, the effect of variations in systemic iron status at perimenopause with respect to both variations in brain iron and variations in cognitive performance and brain function is unknown. This study aimed to assess the relationship between variations in systemic iron at the menopausal transition with (a) behavioral and concurrent electroencephalographic (EEG) measures of cognitive performance and associated brain function and (b) magnetic resonance imaging (MRI) measures of brain iron deposits.

Consider first the possible outcomes for relationships between systemic iron and cognitive performance and brain function. If a negative relationship is found, the outcome would be consistent with the results of Lam et al. [38] and would suggest that higher iron levels might play a causal role in the “brain fog” reported by women at the transition [41,42]. If no relationship is found, the outcome would be consistent with the results of Andreeva et al. [39] and would suggest that variations in iron levels at the cessation of menstruation have no relationship to concurrent cognitive changes. If a positive relationship is found, it would suggest that the positive regularities relating iron levels, cognitive performance, and EEG measures of brain function in women of reproductive age [17,43,44,45,46,47,48,49,50] hold for women at the menopausal transition. However, this outcome would need to be considered in the context of the relationship involving systemic and brain iron.

If the relationship between systemic iron levels and brain iron levels is either negative or null, this would partially contradict the results noted earlier [34,37], suggesting that possibly addressing low iron levels at the transition might result in positive consequences, particularly if a positive relationship is found between systemic iron levels and cognitive performance. Finally, if the relationship between systemic iron levels and brain iron levels is positive, then this would be partially consistent with the results noted earlier [34,37] and would suggest an unfortunate trade-off should there be a positive relationship between systemic iron levels and cognitive performance.

We should note that the original design for this study was a factorial design based on a crossing of participants’ iron status (at or below the 40th sFt percentile specific to participants’ age and race/ethnicity, within the 3rd quartile) and menopausal status (early vs. late). However, a number of serious challenges to recruitment, including the shutting down of data collection during the pandemic, and a general reticence with coming in for a laboratory after restrictions had eased, meant that our period of funding expired before we could meet our enrollment target. The resultant dataset is suitable for exploring the potential correlations but not for testing our original hypotheses.

## 2. Materials and Methods

### 2.1. Participants

A total of 521 women were initially screened by online questionnaire for eligibility (see Figure 1). Of these, 319 participants were excluded based on inclusion/exclusion criteria (specified below). Common reasons of exclusion included being outside of the target age range, not experiencing symptoms of menopause, current treatment for iron deficiency, and high BMI. A total of 202 participants were initially enrolled and had blood samples taken. After this, an additional 163 participants were excluded either based on iron measures or because they were lost to follow up. A total of 39 participants completed the behavioral testing, although another 13 participants were lost due to follow up or incomplete data. A total of 27 cognitive datasets from the participants and 17 MRI scans will be included, due to incomplete or lost data.

### 2.2. Initial Online Screening

Participants completed an initial screening questionnaire using an online survey, which included questions addressing the participant’s age, self-reported BMI (calculated from self-reported height and weight), English proficiency, iron supplementation use, presence of menopausal symptoms, and presence of depression including current use of medications. To move forward in the study, potential participants had to be experiencing symptoms of menopause, not be pregnant or lactating, not be receiving treatment for depression or iron deficiency, have normal or normal-to-corrected vision, have the ability to speak and read English, and have a BMI between 18 and 30.

### 2.3. Laboratory Tests

Participants underwent a blood draw to determine blood iron levels and menopausal status. Blood samples were obtained by trained phlebotomists at the University of Oklahoma Health Sciences Center (OUHSC). Laboratory tests were performed on these samples in the laboratories of OU Medicine using standard clinical practices. The variables related to iron status and inflammation were hemoglobin (Hb, g/dL), serum ferritin (sFt, ng/mL), mean corpuscular volume (MCV, fL), red blood cell count (RBC, M/mm^3^), C-reactive protein (CRP, mg/L), white blood cell count (WBC, K/mm^3^), hematocrit (HCT, %), mean corpuscular hemoglobin (MCH, pg), mean corpuscular hemoglobin concentration (MCHC, g/dL), and red blood cell distribution width (RDW, %). Follicle-stimulating hormone (FSH, mIU/mL) was measured to assess and confirm menopausal status.

An OB/GYN physician (PM) determined each potential participant’s stage of menopause based on the criteria from the Stages of Reproductive Aging Workshop (STRAW+10) [51], using individual responses on the Menopause Health Questionnaire created by the North American Menopause Society (NAMS) and the level of follicle-stimulating hormone (FSH). Participants were considered to be eligible if they were in either STRAW+10 stages 3b or 3a (early) or stages 1 (a b, c) or 2 (late).

Raw values of sFt were converted to percentiles of age- and race/ethnicity-determined distributions derived from the NHANES datasets (https://www.cdc.gov/nchs/nhanes/index.htm, accessed on 17 February 2025) This measure was taken as sFt values vary as a function of both age and race/ethnicity [52]. NHANES datasets from the years 1999–2018 were combined, and data for women were selected and combined in five-year age bins for the four race/ethnicity categories that were represented in all the datasets: non-Hispanic white, non-Hispanic black, Hispanic, and other. The percentiles of the log-normal distributions for each combination of age bin race/ethnicity category were calculated and saved as a searchable dataset, which allowed for the conversion of raw sFt values to percentiles specific to age and race/ethnicity.

Participants were eligible for the study if they were either at or below the 40th percentile or in the 3rd quartile of the sFt distribution for their race/ethnicity and age. Participants were not eligible if they showed elevated levels of CRP (CRP > 8 mg/L), which can mask iron deficiency by artificially elevating sFt values due to inflammation [53]. Participants with highly elevated levels of sFt, indicating iron overload (possible hemochromatosis, sFt > 150 mg/L), were excluded [54].

### 2.4. Structural MRI

After both menopausal and iron status were determined, participants underwent a structural magnetic resonance imaging (MRI) scan to determine regional estimates of brain iron concentrations. MR images were collected using a gradient-echo based susceptibility-weighted imaging (SWI) sequence using four echo times (TE) of 7.53–30 ms with an inter-echo interval of 7.5 ms, a repetition time (TR) of 35 ms, and a flip angle (FA) of 25° on a 1.5T General Electric (GE) system. The SWI image was derived from the GE-exclusive SWAN (T2* weighted angiography) sequence. Images were acquired with 2 mm in the slice direction, as well as a field-of-view of 2280 × 998 mm, and displayed on a 512 × 512 matrix. The GE sequences acquired included T2 fast spin echo in the axial plane, a fast spoiled gradient-echo rapid sequence in the axial slice, spoiled gradient recalled acquisition in the steady state in the coronal plane, and fast spoiled gradient-echo rapid sequence magnetization transfer imaging in the sagittal and coronal slices. The primary regions of interest (ROIs) for the quantification of iron deposits included the globus pallidus, caudate nucleus, putamen, substantia nigra, ventrolateral prefrontal cortex, and anterior cingulate, given both the differential levels of iron deposits in these regions and the findings that age-related accumulation of iron in these regions has been related to declines in cognitive performance [12,29,30,31,55,56,57]. We should note that level of hydration and time since consumption of any significant source of iron were not controlled for. The former is a potential confound as level of hydration has the potential to affect SWI [58], a point which we return to in the Discussion Section. We do not consider the latter to be a potential confound, given the time required for absorption and transport of iron and the time required for either diet or supplements to affect circulating iron levels [59,60].

### 2.5. Cognitive Testing

At the beginning of this session, the participant’s weight (cm) and height (kg) were measured using a digital scale and stadiometer (respectively) and were used to calculate and confirm BMI (kg/m^2^). Participants then completed four cognitive tasks: a face/name associative memory task (FNAM) [61] with immediate and delayed recall, a probabilistic selection task (PST) [62], a rule-based category learning task [63,64], and a visuospatial working memory task [65]. All tasks were conducted while concurrent EEG data were acquired, and we also recorded five minutes of resting EEG data. The testing session was performed in a dimly lit sound-, light-, and electromagnetically shielded chamber and took approximately 3 h to complete. A Mac Mini running Psychophysics Toolbox for Matlab [66,67,68] was used to present stimuli and collect responses. All stimuli were presented on a 61 cm (diagonal) monitor with a resolution of 1920 × 1080 pixels and a gray-to-gray time of 1 ms. Participants used the computer keyboard to make their responses in the behavioral tasks. Participants sat at a fixed distance of 72 cm from the center of the monitor with their heads placed in a chin rest. Each session began with a five-minute resting period, followed by the FNAM with immediate recall. The remaining three tasks were ordered according to a balanced Latin square, followed by the delayed recall for the FNAM. Next, we present brief descriptions of each of the cognitive tasks. Complete methodological details can be found in the Appendix A.

#### 2.5.1. FNAM

The FNAM assesses episodic memory for the association of faces with names and putative occupations and has been shown to show some discriminative sensitivity to the presence of mild cognitive impairment [61,69,70,71]. The FNAM consisted of three phases: a learning phase, followed by an initial recall phase, and a final delayed recall phase, which occurred approximately two hours after the initial learning and recall phases (at the end of the testing session).

#### 2.5.2. PST

The PST measures positive and negative reinforcement learning, which can be affected by variations in DA levels [62,72]. The PST involved two phases: a training phase and a testing phase. In the training phase, participants were presented with three pairs of stimuli and needed to learn which member of each pair was rewarded more frequently. One member of each pair was rewarded more frequently than the other, and the differential reward rate was probabilistic. After meeting a learning criterion, the test phase involved all possible pairings of stimuli.

#### 2.5.3. RBCL

The RBCL is drawn from a literature examining the hypothesized role of two distinct forms of memory—declarative and procedural—in category learning [73,74,75], with rule-based learning associated with declarative memory and DA playing a role in the maintenance and switching of response rules. Deficits in rule-based learning have been documented in Parkinson’s disease [63,64,76], and we have recently [17] shown that women of reproductive age who are experiencing iron deficiency in the absence of anemia show initial deficits in learning rule-based categorization. The task involved the presentation of grayscale Gabor patches, varying in spatial frequency and orientation, with participants required to make categorization decisions based on the values of these two dimensions.

#### 2.5.4. VSWM

The VSWM was included on the basis of a set of findings suggesting that variations in performance on tests of this form of working memory can be related to variations in DA as well as age [77,78,79,80]. Participants were presented with a circular display of 16 white boxes in which blue (target) and red (distractor) circles were located and participants were told to remember the locations of the targets. They were then tested by way of a probe (a question mark) placed in one of the circles with the requirement to indicate whether or not a target had been present in that location.

### 2.6. Ethics

All procedures were reviewed and approved by the Institutional Review Board of the University of Oklahoma (IRB approval number 9388; date of approval 7 June 2018). Participants provided written informed consent and gave their permission for us to acquire protected health information at the start of their involvement.

### 2.7. Statistical Analysis

Values of sFt and reaction time (RT) were assessed for normality prior to any analyses, and both variables were found to not depart from normality. All correlations were performed as Pearson correlations with *p*-values adjusted according to the false discovery rate. Repeated measures analyses on the behavioral and EEG data were performed according to the factorial structure of the tasks as repeated measures mixed models with the factors of each task as fixed and subjects as random factors, and Tukey tests were used for all post-hoc analyses. All analyses were performed using SAS 9.4 for Linux and R version 4.4.1. The criterion for inferring statistical significance was α<0.05.

## 3. Results

Characteristics of the participants are presented in Table 1. The mean age of the participants was 53.96 years (SD = 5.2), and 92.6% of participants self-reported as Caucasian. The mean sFt was 61.90 ng/mL (SD = 34.12), indicating that participants were well above the research criteria for iron deficiency, while the mean sFt percentile (from the NHANES distribution) was 40.31 (SD = 2.41), with a median of 35.40, indicating that participants’ iron levels were below expectation for their age and race/ethnicity. Mean Hb (g/dL) was 13.66 (SD = 0.90), indicating that overall participants were not anemic (note that one participant was just under the criterion value for anemia at 11.9 g/dL). The mean FSH (mIU/mL) value was 67.43 (SD = 38.21), a value consistent with the fact that the majority of the sample were participants in the late stages of menopause. The mean BMI of the women was 25.2 kg/m^2^ (SD = 2.5), consistent with our exclusion of women who were obese, although it was the case that one participant’s measured (rather than self-reported) BMI was just slightly above criterion (30.2 kg/m^2^).

### 3.1. Behavioral Variables

Here, we focus on the relationships involving the behavioral variables and the iron status biomarkers. The variables from each of the tasks were analyzed according to the factorial structure of the tasks, and given that those outcomes are not pertinent to the questions under consideration, we report the details of those analyses in the Appendix A. We report (Table 2) the magnitude of the correlations for which the associated *p*-values remained below the statistical criterion for significance after adjusting for the false discovery rate. The complete set of correlations and all *p*-values are available in the Appendix A.

In brief, the following were the dependent variables obtained for each task. For the FNAM, we measured overall accuracy for tests of face/name and face/occupation associations, both immediately and after a delay along with RTs for correct responses for both associations at both test times. In addition, because the delayed test included new faces, we were able to calculate hit rates (correctly identifying a face as old) and false alarm rates (incorrectly identifying a new face as old), and from these, we were able to calculate the signal detection theory measures of discriminability ($d^{\prime }$) and response criterion (*c*) [81,82]. For the PST, we calculated accuracy for choosing the most-rewarded element of a test pair (referred to as choose A) and the accuracy for avoiding the least-rewarded element (referred to as avoid B). We also calculated RT for the correct choice on trials for which the learned probabilities were similar (referred to as high-conflict trials) and trials for which the probabilities were different (referred to as low-conflict trials). For the RBCL, in each block of 200 trials, we calculated the overall accuracy of the categorization judgment. In addition, due to the nature of the response assignment, in each block of 200 trials, we were able to calculate the marginal hit and false alarm rates for orientation and spatial frequency, and from those, we also calculated values of $d^{\prime }$ and *c* (we refer to these as the marginal measures). Finally, for the VSWM, we were able to calculate each of the following at each level of the task design: overall accuracy, RT for correct responses, and hit and false alarm rates. From the latter, we calculated $d^{\prime }$, *c*, *K* (a measure of working memory capacity, [83]), as well as a modified version, $K^{\prime }$.

The iron status biomarker that was most frequently associated with the behavioral variables was sFt percentile: the value of sFt expressed as the percentile for the participant’s age and race ethnicity. In all cases, higher values of sFt percentile were associated with higher levels of performance: higher accuracy, higher discriminability (as measured using the signal detection measure $d^{\prime }$ [81,82], and shorter RTs. The same types of relationships were observed for raw sFt values but not as frequently as for the sFt percentile. It should be noted that none of the participants met the research criterion for iron deficiency (<either 12 or 15 ng/mL). and that there were only three participants in the sample whose sFt levels would even be consistent with a research definition of diminished iron levels (<20 ng/mL).

It was also the case that the variables indicative of oxygen transport capability—Hb, RBC, HCT, MCV, MCH, MCHC, and RDW—were also related to performance, with there being at least one significant correlation with these variables in each task. In all but a small set of cases, better iron status was associated with better performance. This was true in a sample in which only one participant met the criterion for anemia, and in that case, Hb was quite close to the criterion.

Finally, we should note that significant correlations with age were obtained only for the VSWM. In all cases for this task, higher age was associated with lower performance. The fact that we failed to observe this relationship in any of the other tasks is most likely due to this being a sample with a restricted age range. In addition, there would be no a priori reason to suspect that this task would be more sensitive to age-related declines in performance than any of the others.

Consequently, it appears that in this sample of perimenopausal women, we observed the same relationships between iron status biomarkers and cognitive performance as has been observed in women of reproductive age, with this being true for women who were neither iron deficient nor anemic but who, for the most part, were below expected levels of sFt for their age and race/ethnicity. Hence, it becomes important to examine the relationships between the iron variables and measures of brain function and brain iron deposits.

### 3.2. EEG Variables

Details concerning the pre-processing of the EEG data, as well as the extraction of the task-specific features, can be found in the Appendix A. Here, as with case of the behavioral variables, we concern ourselves with the potential relationships involving the EEG features and the iron status biomarkers.

In brief, the features extracted for the tasks were as follows. For the FNAM, peak amplitudes were extracted for three time intervals, 200–400 (positive), 400–600 (negative), and 600–800 ms (positive), following the onset of the test stimulus. Values were extracted at three electrodes (11, 62, 129, and 75) matching those used by Guo et al. [84] for face/name and face/occupation associations that were either correctly remembered or forgotten at both the immediate and delayed test. For the PST, three response-locked and two feedback-locked features were extracted, with the response-locked features being the error-related negativity (ERN), error-related positivity (Pe), and correct related negativity (CRN), while the two feedback-locked features were the feedback-related negativity (FRN) for correct and incorrect choices in the testing phase [72]. For the RBCL, the features we extracted were the P300 for correct and incorrect category judgments, the late positive slow wave for correct and incorrect judgments, and the fractional area latency for that late positive slow wave [85]. Finally, for the VSWM, we extracted the P300 at levels of the design at three electrode sites (11, 55, and 62) [80].

Table 3 lists the correlations that remained significant after correction for the false discovery rate. The first thing to note in these results is that, as was true for the behavioral data, the majority of the significant correlations involved sFt, the sFt percentile, or both. For all of these correlations, the sign of the correlation coefficient was consistent with the sign of the component amplitude, indicating that higher levels of sFt (measured either way) were consistent with higher-amplitude signals produced by larger populations of similarly oriented neurons firing synchronously. As was also true for the behavioral data, there were significant correlations between the EEG features and the iron biomarkers associated with oxygen transport, although these were not as numerous as was the case for the behavioral data. In addition, the interpretation of the majority of these correlations is consistent with the interpretation of the correlations involving sFt, with better iron status being associated with stronger EEG signals. Again, we should point out that these outcomes were obtained in women who were neither iron deficient nor anemic (with the one exception being a participant who was very mildly anemic).

Finally, there was at least one significant correlation involving age and an EEG feature for each one of the tasks. In all but two cases (both in the VSWM), the correlations were such that older age was associated with attenuated amplitudes, both positive and negative, consistent with age-related findings in the neurocognitive literature [86].

Consequently, with respect to the effects of variations in iron status on EEG-based measures of brain function, the pattern of results observed here is similar to the patterns observed in women of reproductive age. Critically, there appears to be no evidence that higher levels of iron are associated with any impairments in brain function.

### 3.3. Blink Rates

There have been numerous suggestions that blink rates, both spontaneous and task-related, may be indicative of central DA levels [87,88,89], which, in animal models, have been shown to be affected by iron levels. Indeed, Lozoff and colleagues [90,91] measured spontaneous blink rates in iron deficient and anemic infants and found blink rates to be positively related to iron levels. Consequently, we extracted both spontaneous and task-related blink rates from the EEG data; details are provided in the Appendix A.

Table 4 presents those correlations that remained significant after correcting for the false discovery rate. The complete set of correlations can be found in the Appendix A. None of the correlations involving spontaneous blink rates were significant. However, all of the task-related blink rates significantly related to both sFt and sFt percentile and the blink rates from the FNAM and RBCL were also significantly related to Hb and either RDW or MCHC. In all cases, the sign of the correlation indicates that better iron status was associated with lower task-related blink rates. Although this is the opposite of what we expected, based on previous work [90,91], including our own [17,92], two things should be noted. First, there is some evidence that blink rates are reduced when an individual engages in cognitive processing [93,94], which would be consistent with the facts that participants were engaged in cognitive work and those with better iron status performed at higher levels. Second, several recent papers have called into question the extent to which blink rates are actually indicative of central DA [95,96,97]. Consequently, the conclusion to be (cautiously) drawn from the blink rate data is that higher iron status is associated with greater task engagement.

### 3.4. MRI Estimates of Brain Iron Deposits

The MRI data were used to estimate the levels of iron deposits in six regions of interest in each of the two hemispheres: the caudate, putamen, globus pallidus, substantia nigra, ventrolateral prefrontal cortex, and anterior cingulate cortex. These regions were selected based on previous data showing higher levels of iron in these regions relative to other brain regions [12,29,55,98,99,100,101]. Data for how these estimates were obtained, along with the complete set of correlations with age and the blood iron biomarkers, are available in the Appendix A. After correcting for the false discovery rate, none of the correlations reached the criterion for significance. Thus, we conclude from these data that higher iron status had no relationship to brain iron levels.

## 4. Discussion

We noted at the outset that, while there is compelling evidence for age-related accumulation of brain iron, the literature on the relationships among blood iron levels, brain iron levels, and cognitive performance is quite ambiguous. In addition, evidence regarding these potential relationships in women at the menopausal transition is quite limited. As such, we sought to address these issues in a cross-sectional study involving women at the earliest and latest stages of menopause whose iron levels were either low (at or below the fortieth percentile of the distribution of sFt specific to each participant’s age and race/ethnicity) or not (within the third quartile). Along with measuring a set of iron status biomarkers, we obtained MRI scans to estimate brain iron levels, measured behavioral performance on a set of cognitive tasks, and obtained concurrent EEG data to assess brain function.

We sought to quantify two sets of relationships. The first involved systemic iron status and cognitive performance. Across a set of four tasks, we found that cognitive performance was regularly and positively related to iron status. This means that, among a set of women who were neither iron deficient nor (with one exception) anemic, the relationships between measures of cognitive performance and measures of iron status were the same as those that have been observed regularly in women of reproductive age [44,45,46,47,102]. In addition, we found that higher iron status was associated with higher amplitude features of the EEG, also consistent with previous work with women of reproductive age [17,48,49,50,92], along with reduced task-related blink rates that themselves may be indicative of higher cognitive engagement (though we did note that this latter finding needs to be interpreted with some caution). In sum, these results contradict the results from the two relevant extant studies that have reported results specific to women before and after menopause [38,39].

We next needed to determine whether this positive relationship between blood iron levels and cognitive performance exists in the context of a potentially positive relationship between blood iron levels and levels of brain iron. That possibility would have suggested an unfortunate trade-off between cognitive performance and a risk of increased oxidative stress in brain tissue and subsequent neurodegenerative disease. What we found, however, was that there was no relationship between variations in any of the blood measures of iron status and levels of iron deposited in six brain regions. These results suggest that lower levels of iron status might be functional in the “brain fog” commonly reported by women undergoing the menopausal transition and that if a woman is at or below expected levels of iron when she undergoes the transition, repleting her iron levels might not come at the cost of an increased risk of brain oxidative stress. Of course, before this possibility can be considered, additional, preferably longitudinal, studies are needed. However, we find these, though admittedly preliminary, data to be suggestive of the need for such work. Other issues that should be considered in future include iron levels and both normally changing and perturbed levels of circulating hormones [103].

The strengths of the present study include a specific focus on women at the menopausal transition; quantification of participants’ iron status with respect to what would be expected to be true according to their age and race/ethnicity; the novel combination of measures of blood, cognition, brain function, and brain iron deposits; and the involvement of an OB/GYN with a specialization in treating women at the menopausal transition. That being said, there are notable weaknesses, the most obvious being the limited sample size. It was the case that data collection was disrupted by the pandemic shutdown and was further hindered by a reticence to participate in an in-person laboratory study after the shutdown. It was also the case that our upper limit on BMI was too strict for our environment, as the incidence of obesity in Oklahoma is the third highest in the United States, according to the Centers for Disease Control and Prevention [104], resulting in a high exclusion rate. Finally, we did not control for the level of hydration at the time at which the structural MRI data were acquired, which could represent a potential confound [58].

## 5. Conclusions

In a largely understudied group—women at the menopausal transition—we found positive relationships between levels of systemic iron and cognitive performance and brain function, and this positive relationship was observed in the context of no relationship between blood iron levels and brain iron levels. Essentially, for women at the menopausal transition, better iron status was associated with better cognitive performance and better brain function, without an associated cost with respect to an enhanced risk for neurodegenerative disease (by way of increased oxidative stress). For a number of reasons, these conclusions need to be followed up with larger sample sizes and longitudinal designs, but the outcomes are consistent enough to motivate such work.

## Figures and Tables

**Figure 1 nutrients-17-00745-f001:**
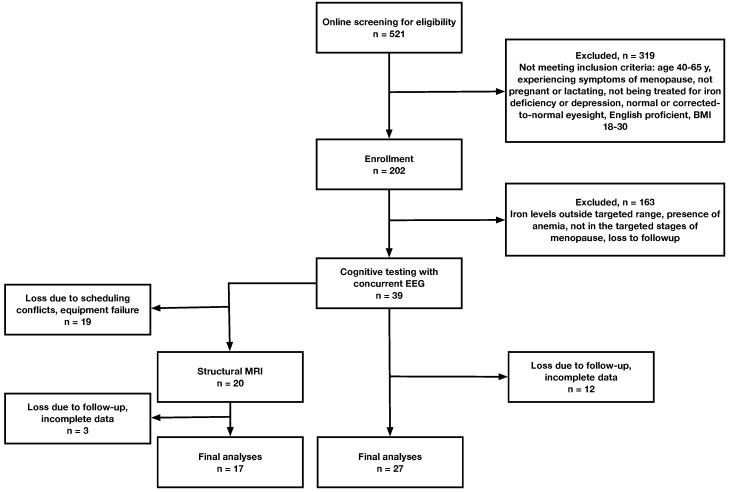
Flow of participants through the study.

**Table 1 nutrients-17-00745-t001:** Characteristics of the participants.

	Mean	SD	Min	Max
Hemoglobin (Hb), g/dL	13.66	0.90	11.90	15.40
Serum ferritin (sFt), ng/mL	61.90	34.12	10.80	135.00
Serum ferritin percentile, NHANES distribution	40.31	2.41	1.20	75.80
C-reactive protein (CRP), mg/L	2.27	1.51	0.10	6.00
White blood cell count (WBC), M/mm^3^	5.87	1.48	3.67	10.19
Red blood cell count (RBC), M/mm^3^	4.53	0.33	3.79	5.56
Hematocrit (HCT), %	41.79	2.41	37.30	47.50
Mean corpuscular volume (MCV), fL	92.51	4.22	82.20	100.50
Mean corpuscular hemoglobin (MCH), pg	30.23	1.49	26.20	33.50
Mean corpuscular hemoglobin concentration (MCHC), g/dL	32.69	0.81	31.30	34.80
Red blood cell distribution width (RDW), %	12.64	0.68	11.60	14.70
Age, y	53.96	5.20	44.00	64.00
FSH, mIU/mL	67.43	38.21	2.00	131.00
BMI, kg/m^2^	25.2	2.8	20.9	30.2
White, N (%)	25 (92.6)			
Early stage menopause, N (%)	8 (29.6)			
Inflammation, N (%)	2 (7.4)			

**Table 2 nutrients-17-00745-t002:** Significant correlations involving the behavioral variables and the iron status biomarkers. All correlations were significant after correcting for false discovery rate. Note: pctl = percentile, targ. = target, and dist. = distractor.

Task	Variable	Age	Hb	sFt	sFt pctl	RBC	HCT	MCV	MCH	MCHC	RDW
FNAM	P(C), immediate face/name				0.46						−0.52
	P(C), delayed face/name			0.47	0.44					0.53	−0.45
	Hit rate, delayed recognition		0.46	0.48	0.48					0.53	−0.52
	$d^{\prime }$, delayed recognition			0.49	0.46					0.51	−0.44
	*c*, delayed recognition		−0.47								0.45
	RT, immediate face/name			−0.43	−0.47						
	RT, delayed face/name			−0.51	−0.59						0.43
	RT, immediate face/occupation				−0.43	0.44		−0.42	−0.49		
	RT, delayed face/occupation		−0.48				−0.46				
PST	P(C), choose A		0.40	0.46							
	P(C), avoid B				0.41						
	RT, high conflict							−0.44	−0.54		
RBCL	P(C), block 2			0.51	0.52						
	Marginal $d^{\prime }$, orientation, block 1									0.43	
	Hit rate, orientation, block 2			0.5	0.42						
	Marginal c, orientation, block 2			−0.42							
	Hit rate, spatial frequency, block 2			0.56	0.59						
	Marginal $d^{\prime }$, spatial frequency, block 2				0.46						
	Marginal *c*, spatial frequency, block 2			−0.58	−0.51						
VSWM	P(C), 5 targ., 0 dist., target absent	−0.49									
	P(C), 5 targ., 0 dist., targ. present			0.51	0.51						
	P(C), 5 targ., 2 dist., targ. present			0.62	0.66		−0.40				
	RT, 3 targ., 0 dist., targ. absent	0.54			−0.65						
	RT, 3 targ., 0 dist., targ. present	0.55			−0.54						
	RT, 3 targ., 2 dist., targ. absent	0.62			−0.68						
	RT, 3 targ., 2 dist., targ. present	0.48			−0.58						
	RT, 5 targ., 0 dist., targ. absent	0.61			−0.58						
	RT, 5 targ., 0 dist., targ. present	0.45			−0.49						0.41
	RT, 5 targ., 2 dist., targ. absent	0.71			−0.47						
	RT, 5 targ., 2 dist., targ. present	0.53									0.41
	Hit rate, 5 targ., 0 dist.			0.54	0.52						
	Hit rate, 5 targ., 2 dist.			0.62	0.66						−0.40
	False alarm rate, 5 targ., 0 dist.	0.49									
	$d^{\prime }$, 5 targ., 0 dist.			0.45	0.52						
	$d^{\prime }$, 5 targ., 2 dist.			0.45	0.52						
	*c*, 3 targ, 0 dist.									−0.41	
	*c*, 5 targ., 0 dist.			−0.43							
	*c*, 5 targ., 2 dist.			−0.65	−0.65						
	*K*, 5 targ., 0 dist.			0.56	0.57						0.45
	*K*, 5 targ., 2 dist.			0.63	0.67						
	$K^{\prime }$, 5 targ., 0 dist.			0.49	0.49						
	$K^{\prime }$, 5 targ., 2 dist.			0.51	0.60						

**Table 3 nutrients-17-00745-t003:** Significant correlations involving the EEG variables and the iron status biomarkers. All correlations were significant after correcting for false discovery rate.

Task	Variable	Age	Hb	sFt	sFt pctl	RBC	HCT	MCV	MCH	MCHC	RDW
FNAM ^1^	immed., 200–400, Fz, remem., pos. amp.			0.60							
face/name	immed., 200–400, Cz, remem., pos. amp.			0.69	0.50						
	immed., 200–400, Oz, forgot., pos. amp			0.55	0.69						
	immed., 400–600, Fz, remem., neg. amp.			−0.52	−0.69						
	immed., 400–600, Cz, remem., neg. amp.			−0.62	−0.69						
	immed., 400–600, Oz, forgot., neg. amp.			−0.59							
	immed., 600–800, Fz, remem., pos. amp			0.66	0.55						
	delay., 200–400, Pz, forgot., pos. amp.				−0.49						
	delay., 200–400, Pz, remem., pos. amp.			0.49							
	delay., 200–400, Oz, forgot, pos. amp.			0.70	0.67						
	delay., 200–400, Oz, remem., pos. amp.			0.51	0.42						
	delay., 400–600, Oz, remem., neg. amp.			−0.78	−0.74						
face/occn	immed., 400–600, Fz, remem., neg. amp.									−0.54	
	immed., 400–600, Pz, remem., neg. amp.									−0.57	
	immed., 600–800, Fz, remem., pos. amp.									0.65	
	immed., 600–800, Fz, forgot., pos. amp.				0.53						
	imeed., 600–800, Pz, forgot., pos. amp.			0.66	0.60						
	immed., 600–800, Oz, forgot, pos. amp.			0.77	0.76						
	delay, 200–400, Cz, remem., pos. amp		0.48								
	delay., 400–600, Fz, remem., neg. amp.	0.48									
	delay., 400–600, Cz, forgot., neg. amp.			−0.49						−0.50	
	delay., 400–600, Cz, remem., neg. amp		−0.53								
	delay., 400–600, Pz, forgot., neg. amp.			−0.63	−0.55						
PST	Error related negativity (ERN)			−0.81	−0.88						
	Error related positivity (Pe)			0.69	0.73						
	Correct related negativity			−0.84	−0.94						
	Feedback related negativity, correct	0.45		−0.75	−0.91						
	Feedback related negativity, incorrect	0.44		−0.76	−0.92						
RBCL	P300, correct	−0.56		0.75	0.94						
	P300, incorrect	−0.57		0.75	0.94						
	Late positive slow wave, correct	−0.55		0.74	0.90						−0.52
	Late positive slow wave, incorrect	−0.59		0.74	0.92						−0.51
	Fractional area latency (ms), correct			0.59	0.48				0.45		
	Fractional area latency (ms), incorrect			0.66	0.75						
VSWM ^2^	P300, 3 targ., 0 dist., targ. abs., E11	0.49									
	P300, 3 targ., 0 dist., targ. pres., E11			0.58							
	P300, 3 targ., 0 dist., targ. pres., E62								0.47	0.47	
	P300, 3 targ., 2 dist., targ.abs., E11			0.48							
	P300, 3 targ., 2 dist., targ. pres., E11			0.48							
	P300, 5 targ., 0 dist., targ. pres., E11			0.45							
	P300, 5 targ., 2 dist., targ. abs., E11								0.47	0.52	
	P300, 5 targ., 2 dist., targ. abs., E55									0.46	
	P300, 5 targ., 2 dist., targ. abs., E62									0.51	
	P300, 5 targ., 2 dist., targ. pres., E11			0.51							
	P300, 5 targ., 2 dist., targ. pres., E55	0.56									

^1^ Abbreviations: immed. = immediate; delay. = delayed; remem. = remembered; forgot. = forgotten; pos. = positive; neg. = negative; amp. = amplitude. ^2^ Abbreviations: targ. = target; dist. = distractors; pres. = present; abs. = absent.

**Table 4 nutrients-17-00745-t004:** Correlations involving task-related blink rates and the iron status biomarkers. All correlations were significant after correcting for false discovery rate.

	Hb	sFt	sFt pctl	MCHC	RDW
FNAM	−0.49	−0.57	−0.60		0.51
PST		−0.47	−0.47		
RBCL	−0.50	−0.43	−0.45	−0.47	
VSWM		−0.53	−0.54		

## Data Availability

All data and materials are available upon reasonable request to the corresponding author.

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
