# Peer review of "Cognitive Performance in Relation to Systemic and Brain Iron at Perimenopause"

_nutrients, 2025, doi:10.3390/nu17050745_

Round 1
Reviewer 1 Report
Comments and Suggestions for Authors The main question addressed by the research is the relationships among iron levels and cognitive performance in women at the menopausal transition.
The topic is original to the field.
It adds to the subject area the fact that low iron levels at the menopausal transition may be a candidate approach for alleviating the “brain fog” commonly experienced at menopause.
Regarding the methodology, it should be added the fact that according to published data, women <43 with a history of bilateral oophorectomy without estrogen replacement therapy may be more susceptible to cognitive deterioration. Thus, the findings of the manuscript could be associated to hormone relations that should be
commented.
The conclusions are consistent with the evidence and arguments presented.
The references are appropriate.
Author Response
We thank the reviewer for the comments and suggestions.
Comment: Regarding the methodology, it should be added the fact that according to published data, women <43 with a history of bilateral oophorectomy without estrogen replacement therapy may be more susceptible to cognitive deterioration. Thus, the findings of the manuscript could be associated to hormone relations that should be commented.
Reply: We have acknowledged this issue, provided a supporting reference and noted it as a issue to be addressed in future work.
Reviewer 2 Report
Comments and Suggestions for Authors
To the Authors
The present study evaluates in a small population of women at the menopausal transition an association between iron status, cognitive performance and brain function.The Authors measured a set of iron status biomarkers, MRI scans to estimate brain iron levels, behavioral performance on a set of cognitive tasks (behavioral tests: FNAM, PST, RBCL, VSWM), and concurrent EEG. The present study conclusion is that higher iron status is associated with better cognitive performance in a sample of women at the menopausal transition who were neither iron deficient nor anemic. The Authors suggest that addressing low iron levels at the menopausal transition may be a candidate approach for alleviating the “brain fog” commonly experienced at menopause. The Ms is original and overall well-planned. A small population size is the key limitation. Study presentation can be improved.
Minor issues
1. LL 314: (Recall that none of the participants met research criterion for iron deficiency). Please replace the colloquial form with a more formal one.
2. LL 320: (An all but a small set of cases, better iron status was associated with better performance). “An all but” small sounds weird/odd. Is it actually “An all but…” to be intended as “In all but…”? Please either correct or clarify.
3. LL 335: (…relationships between the iron variables and our measures of brain function and brain iron) - The Authors could easily omit “our”.
4. LL 380-382: (Indeed, Lozoff and colleagues [87,88] measured spontaneous blink rates in iron deficient and anemic infants and found blink rates to be positively related to blink rates). The statement “found blink rates to be positively related to blink rates “ is nonsense. Please, either correct or clarify.
5. Table 4. (Correlations involving task-related blink rates and the iron status biomarkers. All correlations were signficant after correcting for false discovery rate). The typo (e.g. “signficant”) needs correction.
6. Text. “…along reduced task-related blink rates..”: Along should likely be read as “along with”. Please either correct or clarify.
Author Response
We thank the reviewer for the thoughtful critiques and suggestions.
Comment: LL 314: (Recall that none of the participants met research criterion for iron deficiency). Please replace the colloquial form with a more formal one.
Reply: We have added the numeric criteria for iron deficiency that have been used in the literature.
Comment: LL 320: (An all but a small set of cases, better iron status was associated with better performance). “An all but” small sounds weird/odd. Is it actually “An all but…” to be intended as “In all but…”? Please either correct or clarify.
Reply: Corrected to read "In all but a small set of cases ..."
Comment: (…relationships between the iron variables and our measures of brain function and brain iron)
Reply: Modified as requested.
Comment: The statement “found blink rates to be positively related to blink rates “ is nonsense. Please, either correct or clarify.
Reply: Corrected to read "... and found blink rates to be positively related to iron levels."
Comment: The typo (e.g. “signficant”) needs correction.
Reply: Corrected as requested.
Comment: “…along reduced task-related blink rates..”: Along should likely be read as “along with”.
Reply: Corrected as requested.
Reviewer 3 Report
Comments and Suggestions for Authors
The manuscript by Barnett and collaborators describes a cross-sectional study aimed to assess the relationship between changes in iron status in women under menopausal transition with various measures of cognitive performance, electroencephalographic amplitude features, and intracerebral iron deposits estimated by structural magnetic resonance imaging. Despite the small number of subjects investigated, there is not much literature on the relationship between iron status and cognition in perimenopausal women, which makes this study potentially interesting. The findings indicate that a better iron status was associated with better cognitive performance. Therefore, the authors conclude that addressing a deficient iron status may alleviate cognitive impairment in menopausal women. Statistical methodology, data presentation, number of references, and language are excellent. I will limit myself to a few remarks.
Major remarks
Page 2, line 47. In reality, in the Fenton reaction, it is instead the Fe+2 that reacts with the hydrogen peroxide, producing hydroxyl radicals (Fe2+ + Hâ‚‚Oâ‚‚ → Fe3+ + OH- + •OH)
Page 3, lines 86-90. In my opinion, the results of the study by Andreeva et al. (ref. no. 39) were presented in a slightly biased manner. In perimenopausal women aged ≥ 46y at baseline, there was a slight evidence of a protective effect of lower iron stores (reflected in serum ferritin) in overall cognitive performance and short-term memory.
Page 11, lines 381-382. "and found blink rates to be positively related to blink rates”. I don't understand this sentence.
Minor remarks
The text is quite readable and contains only a few language inaccuracies. Although I do not consider myself qualified, I would like to point out:
Abstract (line 3) “The need to better understand…”
Page 7, line 276. Please add the measurement unit 25.2 kg/m²
Page 11, line 371. The expression “was associated” was repeated twice.
Author Response
We thank the reviewer for the thoughtful comments and suggestions.
Comment: In reality, in the Fenton reaction, it is instead the Fe+2 that reacts with the hydrogen peroxide, producing hydroxyl radicals.
Response: Corrected as requested.
Comment: Page 3, lines 86-90. In my opinion, the results of the study by Andreeva et al. (ref. no. 39) were presented in a slightly biased manner. In perimenopausal women aged ≥ 46y at baseline, there was a slight evidence of a protective effect of lower iron stores (reflected in serum ferritin) in overall cognitive performance and short-term memory.
Response: Corrected to reflect the single significant result in Table 2 of the Andreeva et al. paper.
Comment: Page 11, lines 381-382. "and found blink rates to be positively related to blink rates”. I don't understand this sentence.
Response: Corrected to read "found blink rates to be positively related to iron levels."
Comment: Abstract (line 3) “The need to better understand…”
Response: Rewritten to be more clear as "The need to better understand these potential relationships in women for whom monthly blood loss (and thus iron loss) is ceasing is highlighted by iron's accumulation in brain tissue over time, thought to be a factor in the development of neurodegenerative disease."
Comment: Page 7, line 276. Please add the measurement unit 25.2 kg/m²
Response: Units added as requested.
Comment: Page 11, line 371. The expression “was associated” was repeated twice.
Response: Edited to remove the repetition.
Reviewer 4 Report
Comments and Suggestions for Authors
Journal: Nutrients (ISSN 2072-6643)
Manuscript ID: nutrients-3472592
Type:Article
Title: Cognitive Performance in Relation to Systemic and Brain Iron at Perimenopause
Taking everything considered, this research makes a significant addition to the fields of neuropharmacology and cognitive aging. This work is especially innovative and significant because of the careful participant selection, sophisticated imaging methods, and thoughtfully created cognitive tests. This study has the potential to stimulate more research into the function of iron metabolism in cognitive health because of its robust statistical base and obvious methodological rigor.
Minor revision needed
Abstract:
1. The title highlights both systemic and brain iron in relation to cognitive function during perimenopause, while the abstract indicates that no association was discovered between blood iron and brain iron. This weakens the interaction with brain iron. Consider explaining why brain iron was tested despite the null results.
2. Perimenopause is characterized by variable hormone and iron levels. Were women in early and late perimenopause evaluated separately?
Introduction:
3. (lines 28–30, 93–99, 114–121). Give a precise definition of perimenopause based on accepted clinical standards (e.g., STRAW+10).
Discuss about the potential effects of estrogen changes on brain and systemic iron levels.
Give a rationale for using EEG to measure cognitive function.
4. Although brain iron accumulation is mentioned in the beginning as a risk factor for neurodegeneration, there is no link made between this and cognitive function in healthy perimenopausal women.
Though it does not speculate as to why brain iron may (or may not) be related to cognition in this particular demographic, the study intends to investigate both systemic and brain iron.
Describe how brain iron is predicted to affect perimenopausal cognitive function, not just neurodegeneration.
Give a more convincing explanation for the study's measurement of brain iron.
Participant Selection and Screening
5. How did you choose the third quartile and 40th percentile as the proper cutoff points for the sFt percentile when choosing participants? Did previous research or exploratory analysis serve as the basis for these thresholds?
MRI and Cognitive Testing
6. Were there any possible confounding factors that could affect susceptibility-weighted imaging results in MRI-based iron quantification, such as hydration state or recent dietary iron intake?
Statistical Analysis
7.Was the relationship between iron levels and menopausal stage and cognitive performance studied in conjunction, or separately?
Author Response
We thank the reviewer for the careful review and thoughtful suggestions.
Comment: Abstract: Consider explaining why brain iron was tested despite the null results.
Response: This was a finding from the study, it was not known in advance. The data had to be examined before we could know anything about the possible relationship.
Comment: Perimenopause is characterized by variable hormone and iron levels. Were women in early and late perimenopause evaluated separately?
Response: Unfortunately, the eventual sample size was too small to allow this to be done.
Comment: (Introduction) (lines 28–30, 93–99, 114–121). Give a precise definition of perimenopause based on accepted clinical standards (e.g., STRAW+10). Discuss about the potential effects of estrogen changes on brain and systemic iron levels. Give a rationale for using EEG to measure cognitive function.
Response: The specific STRAW+10 stages considered for inclusion are now specified in the methods section. The need to consider possible interactions between iron and hormone levels is now noted in the Discussion. The rationale for including EEG has been bolstered and the previously-confusing statement about using EEG for measuring cognitive function as been clarified to read "... behavioral and concurrent electroencephalographic (EEG) measures of cognitive performance and associated brain function"
Comment: Give a more convincing explanation for the study's measurement of brain iron.
Response: We should respectfully that we have dedicated five pararaphs to the need to measure brain iron and have, in this revision, selected added emphases in selected locations to make the need for these measurements more apparent.
Comment: How did you choose the third quartile and 40th percentile as the proper cutoff points for the sFt percentile when choosing participants? Did previous research or exploratory analysis serve as the basis for these thresholds?
Response: These levels were chosen from statistical reasoning to allow for two sets of participants with non-overlapping iron levels.
Comment: Were there any possible confounding factors that could affect susceptibility-weighted imaging results in MRI-based iron quantification, such as hydration state or recent dietary iron intake?
Response: We note in the section on the MRI methods that we did not control for hydration or iron intake, and we return to the potential confound posed by not controlling for hydration in the discussion.
Comment: 7.Was the relationship between iron levels and menopausal stage and cognitive performance studied in conjunction, or separately?
Response: The limited sample size and the low representation of women at the earliest stages prevented us from making any comparisons as a function of menopausal stage.